# Learning Quadruped Locomotion Policies using Logical Rules

**Primary Keywords:** *(2) Learning; (3) Robotics*

## Abstract

Quadruped animals are capable of exhibiting a diverse range of locomotion gaits. While progress has been made in demonstrating such gaits on robots, current methods rely on motion priors, dynamics models, or other forms of extensive manual efforts. People can use natural language to describe dance moves. Could one use a formal language to specify quadruped gaits? To this end, we aim to enable easy gait specification and efficient policy learning. Leveraging Reward Machines (RMs) for high-level gait specification over foot contacts, our approach is called RM-based Locomotion Learning (RMLL), and supports adjusting gait frequency at execution time. Gait specification is enabled through the use of a few logical rules per gait (e.g., alternate between moving front feet and back feet) and does not require labor-intensive motion priors. Experimental results in simulation highlight the diversity of learned gaits (including two novel gaits), their energy consumption and stability across different terrains, and the superior sample-efficiency when compared to baselines. We also demonstrate these learned policies with a real quadruped robot.

## 1 Introduction

Legged animals are capable of performing a variety of locomotion gaits, in order to move efficiently and robustly at different speeds and environments (Hoyt and Taylor 1981). The same can be said of legged robots, where different locomotion gaits have been shown to minimize energy consumption at different speeds and environments (Fu et al. 2021; Da et al. 2021; Yang et al. 2022). Still, leveraging the full diversity of possible locomotion gaits has not been thoroughly explored. A larger variety of gaits can potentially expand a quadruped's locomotion skills or even enable traversing terrains that are not possible before. Unfortunately, learning *specific* quadruped locomotion gaits is a challenging problem. To accomplish this, it is necessary to design a reward function which can express the desired behavior. Commonly used reward functions for quadruped locomotion encourage maximizing velocity command tracking, while minimizing energy consumption (Tan et al. 2018; Kumar et al. 2021). While training over these types of reward functions oftentimes yields high quality locomotion policies, they do not specify any particular gait.

In order to incentivize the agent to learn a specific gait, the reward function must be encoded with such gait-specific knowledge. It is possible to design a naive reward function which explicitly encourages specific sequences of milestone foot contacts, which we refer to as poses. Unfortunately, this breaks the Markov property, because historical knowledge of previous poses within the gait is necessary to know which pose should be reached next in order to adhere to the specified gait. Quadruped locomotion controllers are commonly run at 50 Hz or more (Kumar et al. 2021; Miki et al. 2022), which generates a long history of states between each pose of a gait. Thus, naively satisfying the Markov property would require including all of these historical states in the state space, and would make the learning process more challenging as the policy would need to figure out which portion of this history is relevant.

Some researchers have taken advantage of motion priors in order to encode gait-specific knowledge in a reward function. One popular method for encoding such knowledge in a reward function is to maximize the similarity between the robot's motion and a reference trajectory (Peng et al. 2020; Smith et al. 2021). While this approach has been successfully demonstrated on real robots, it requires significant manual effort to obtain reference trajectories, and constrains the robot's motion to the given trajectory.

In this paper, we alleviate the above mentioned problem of gait specification by leveraging Reward Machines (RMs) (Icarte et al. 2022), which specify reward functions through deterministic finite automatons. RMs have been applied to various domains for guiding RL agents (Xu et al. 2020; Neary et al. 2020; Camacho et al. 2021; Dohmen et al. 2022). In this paper, RM serves as high-level specifications of gaits for low-level locomotion policy learning. The RM transition function is defined through Linear Temporal Logic (LTL) formulas over propositional symbols, which in our case specify foot contacts. Thus, changing the automaton state corresponds to reaching the next pose within the gait. The reward function is Markovian when considering the low-level state (robot sensor information), along with the current automaton state, because the automaton state encodes the relevant gait-level information needed to determine the next pose. This approach enables us to easily specify and learn diverse gaits via logical rules, without the use of motion priors.

We refer to our approach as RM-based Locomotion

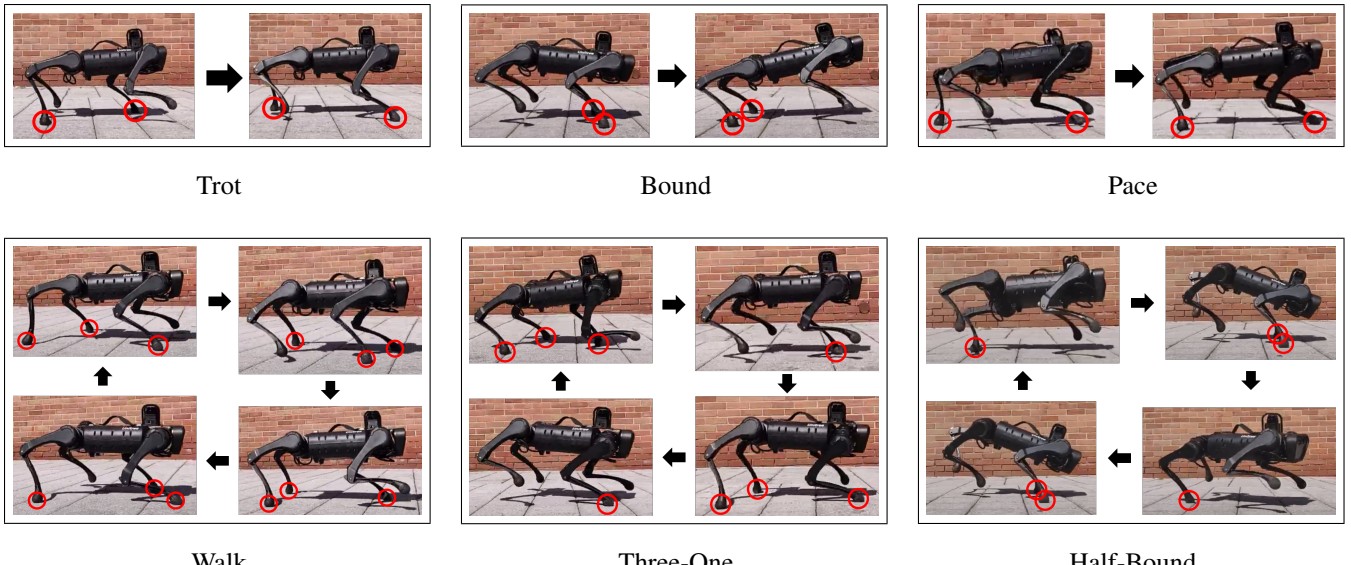

Figure 1: Snapshots of important poses of each of the six gaits learned with six different RMs. Specifying and learning the gaits require defining an automaton with no more than five automaton states (only two for half of the gaits). Red circles are around feet making contact with the ground.

Learning (RMLL), and train policies for six different gaits in simulation without the use of reference trajectories. Each policy is trained over a range of gait frequencies, which we can dynamically adjust during deployment. The reward function of each gait is defined through an automaton over desired foot contacts. We conduct an ablation study to evaluate the sample efficiency of RMLL in training the six different gaits, measure energy consumption and stability of each gait in different terrains, and deploy all gaits on a real Unitree A1 quadruped robot (see Figure 1). We compare RMLL to three baselines, each of which is designed to evaluate whether knowledge of the automaton state during training is actually beneficial in terms of sample efficiency. Results show that RMLL improves sample efficiency over its ablations for all gaits, more substantially for more complex gaits.

## 2 Related Work

In this section, we discuss prior work on RMs, and legged locomotion via Reinforcement Learning (RL). We then focus on existing methods of gait specification and learning for legged locomotion, with and without motion priors.

### 2.1 Reward Machine

Since the introduction of Reward Machines (**RMs**) (Icarte et al. 2018), there have been various new research directions such as learning the RM structure (Xu et al. 2020), RM for partially observable environments (Toro Icarte et al. 2019), multi-agent intention scheduling (Dann et al. 2022), and probabilistic RMs (Dohmen et al. 2022) to name a few. While these works primarily focused on RM algorithmic improvements and theoretical analysis, their applications did not go beyond toy domains. RMs have also been used for simulated robotic arm pick-and-place tasks, which learn RM

structures from demonstrations (Camacho et al. 2021). However, their approach was not implemented or evaluated in real-world robotic continuous control problems with high-dimensional action spaces. A recent journal article formally described the RM framework as well as a few algorithms for RM-based reinforcement learning (Icarte et al. 2022). We use RMs for robot locomotion learning in this work.

### 2.2 RL-based Locomotion Learning

There are numerous works on applications of RL for quadruped robot locomotion policy learning (Tan et al. 2018; Kumar et al. 2021; Smith et al. 2021; Rudin et al. 2022; Miki et al. 2022; Zhuang et al. 2023). Approaches of this type often lead to robust locomotion gaits, some of which can transfer to real robots. However, the above mentioned approaches generally focus on learning robust locomotion policies, and do not support the specification of particular gaits. Other works that support diverse locomotion gaits are described next.

### 2.3 Diverse Locomotion Gaits

**With Motion Priors:** Various works have learned diverse locomotion gaits for quadruped robots with the use of motion priors. For example, trajectory generators (Iscen et al. 2018) and motion references (Smith et al. 2021; Peng et al. 2020) have been leveraged for learning specified gaits. Obtaining these priors require extensive human (and sometimes even animal) effort, and restricts the robot to following the specified trajectory with little variation. While motion references can be generated, it requires highly tuned foot trajectory polynomials and phase generation functions (Shao et al. 2021). Our approach does not require such motion priors and can easily specify different gaits via a few logical

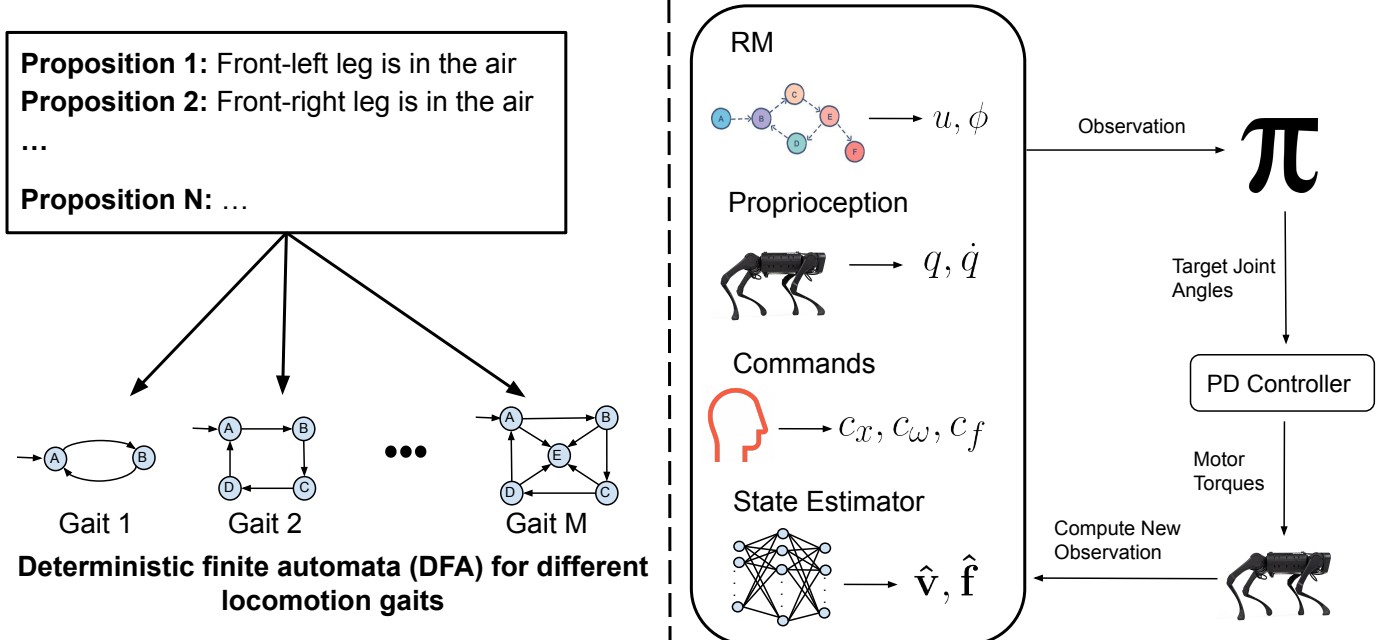

Figure 2: Overview of RM-based Locomotion Learning (RMLL). We consider propositional statements specifying foot contacts. We then construct an automaton via LTL formulas over propositional statements for each locomotion gait (left side). To train gait-specific locomotion policies, we use observations which contain information from the RM, proprioception, velocity and gait frequency commands, and variables from a state estimator (right side).

rules. Our policies also have freedom to explore variations of the specified gait on its own and is not restricted by a predefined trajectory.

**MPC-based:** Various MPC-based approaches have successfully demonstrated diverse locomotion gaits without the use of motion priors (Di Carlo et al. 2018; Kim et al. 2019). However, these methods require accurate dynamics models, and significant manual tuning for each gait.

**Emergent Gaits:** Different gaits can naturally emerge through minimizing energy (Fu et al. 2021), or selected from a high-level policy which selects foot contact configurations or contact schedules (Da et al. 2021; Yang et al. 2022). More generally, diverse exploration strategies have been shown to improve policy performance and encourage learning different behaviors (Cohen, Yu, and Wright 2018). While these approaches lead to diverse locomotion gaits and behaviors, it does not provide the ability to learn any *arbitrary* gait or gait frequency specified beforehand.

**Most Similar to ours:** There are recent works that aim to learn locomotion gaits based on high-level gait descriptions – RMLL (ours) shares the same spirit. LLMs have been leveraged to specify and perform diverse locomotion behaviors (Yu et al. 2023; Tang et al. 2023). While the work of Tang et al. is useful in converting natural language to low-level control on hardware, they require extensive prompt engineering, and additional manual effort in defining a random pattern generator for each desired gait. Thus, this approach

has only been demonstrated on two-beat gaits, and can be less effective to uncommon gaits, while our RMLL approach supports specifying and learning the two novel gaits of Three-One and Half-Bound. Other similar works involve specifying and learning diverse locomotion gaits through explicitly defining swing and stance phases per leg (Siekmann et al. 2020; Margolis and Agrawal 2023). The former approach is designed for a bipedal robot, while the later only supports two-beat quadruped gaits. In contrast, RMLL only needs foot contact sequences (instead of leg-specific timings), and can specify and learn arbitrary quadrupedal gaits well beyond the set of two-beat gaits.

The main contribution of this research is a novel paradigm for gait specification. We focus on demonstrating the complete pipeline of this new paradigm using a real robot. This research paves the way for future research, e.g., on improving the efficiency of policy learning, developing novel gaits for quadrupedal robots, and intelligently transitioning between gaits.

## 3 RM-based Locomotion Learning

We present our RM-based reinforcement learning approach for learning quadruped locomotion policies below. Figure 2 presents an overview of how we use RMs to specify a diverse set of quadruped locomotion gaits and facilitate efficient policy learning.

## 3.1 Reinforcement Learning

RL algorithms expect environments to be modeled as an MDP of form $M = (S, A, T, R, \gamma)$. $S$ refers to the state space, $A$ is the set of actions the agent can take, $T : S \times A \times S \to [0,1]$ is the transition function which outputs the probability of reaching state $s'$ given state $s$ and action $a$, $R : S \times A \times S \to \mathbb{R}$ is the reward received by taking action $a$ from state $s$ and ending up in state $s'$, and $\gamma$ is the discount factor which determines how valuable future reward should be considered in comparison to immediate reward.

The goal in RL is to find a policy $\pi : S \to A$ which selects actions that maximizes expected future discounted reward, given a state. Importantly, the agent does not have knowledge of transition or reward functions, and can only learn through trial and error experiences in the environment.

## 3.2 Reward Machines: Concepts and Terminologies

Reward Machines are typically used in settings where there is a set of "milestone" sub-goals to achieve in order to complete some larger task. Reward functions which do not encode these subgoals are oftentimes too sparse, while reward functions which explicitly reward sub-goal completion can be non-Markovian. An RM allows for specification of these sub-goals through an automaton, which can be leveraged to construct an MDP. Thus, through an RM, the reward function can give positive feedback for completing sub-goals, while also defining an MDP with a Markovian reward function.

Formally, an RM is defined as the tuple $(U, u_0, F, \delta_u, \delta_r)$ (Icarte et al. 2018), where $U$ is the set of automaton states, $u_0$ is the start state, $F$ is the set of accepting states, $\delta_u : U \times 2^{\mathbf{P}} \to U \cup F$ is the automaton transition function, while $\delta_r : U \times 2^{\mathbf{P}} \to [S \times A \times S \to \mathbb{R}]$ is the reward function associated with each automaton transition. This RM definition assumes the existence of set $\mathbf{P}$, which contains propositional symbols that refer to high-level events from the environment that the agent can detect. A labelling function $L : S \times A \times S \to 2^{\mathbf{P}}$ determines the truth values of each symbol in $\mathbf{P}$ at each environment step. Then, the agent evaluates which automaton state transition to take via $\delta_u$, and receives reward via $\delta_r$.

Reward machines are defined alongside state space $S$, which describe the low-level observations the agent receives after each step in the environment. In order to construct an MDP from the non-Markovian reward defined by the RM, the agent considers its own observations from $S$, along with its current RM state from $U$. Training over state space $S \times U$ no longer violates the Markov property, because knowledge of the current RM state indicates which sub-goal was previously completed. The inclusion of this subsection is simply for the completeness of this paper. More details are available in the RM article (Icarte et al. 2022).

## 3.3 RM for Quadruped Locomotion

We use RMs to specify the sequence of foot contacts expected of the gait. In our domain, we consider $\mathbf{P} = \{P_{FL}, P_{FR}, P_{BL}, P_{BR}\}$, where $p \in \mathbf{P}$ is a Boolean

| Term Description | Definition | Scale |
|---|---|---|
| Linear Velocity $x$ | $exp(-\|\mathbf{c_x} - \mathbf{v_x}\|^2/0.25)$ | 1.0 |
| Linear Velocity $z$ | $\mathbf{v_z}^2$ | $-2.0$ |
| Angular Velocity $x, y$ | $\|\omega_{\mathbf{x,y}}\|^2$ | $-0.05$ |
| Angular Velocity $z$ | $exp(-(\mathbf{c}_\omega - \omega_{\mathbf{z}})^2/0.25)$ | 0.5 |
| Joint Torques | $\|\tau\|^2$ | $-0.0002$ |
| Joint Accelerations | $\|(\dot{\mathbf{q}}_{\mathbf{last}} - \dot{\mathbf{q}})/dt\|^2$ | $-2.5e{-}7$ |
| Feet Air Time | $\sum_{f=1}^4 (\mathbf{t_{air,f}} - 0.5)$ | 1.0 |
| Action Rate | $\|\mathbf{a_{last}} - \mathbf{a}\|^2$ | $-0.01$ |

Table 1: All terms of $R_{\mathbf{walk}}$. $\mathbf{v}$ refers to base velocity, $\mathbf{c}$ refers to commanded linear and angular base velocity, $\omega$ refers to base angular velocity, $\tau$ refers to joint torques, $\dot{\mathbf{q}}$ refers to joint velocities, $\mathbf{t_{air}}$ refers to each foots air time, and $\mathbf{a}$ refers to an action.

variable. These indicate whether the front-left (FL), front-right (FR), back-left (BL), and back-right (BR) feet are making contact with the ground. Labelling function $L$ evaluates whether a foot makes contact with the ground via the foot force sensors on the robot. Automaton states in $U$ correspond to different poses in the gait, where $u_0$ corresponds to the last pose. Meanwhile, $\delta_u$ changes the automaton state when the next pose in the gait is reached. We define $\delta_r$ as:

$$\delta_r(u_t, a) = \begin{cases} R_{\mathbf{walk}}(s) * b & \delta_u(u_t, a) \neq u_t \\ R_{\mathbf{walk}}(s) & otherwise \end{cases}$$

where $R_{\mathbf{walk}}$ encourages maximizing velocity command tracking while minimizing energy consumption (Rudin et al. 2022), and is fully defined in Table 1. Reward function $\delta_r$ encourages taking RM transitions which correspond to the specified gait, because $R_{\mathbf{walk}}$ is scaled by bonus $b$ when such transitions occur. We leave $F$ empty for all gaits, as quadruped locomotion is an infinite-horizon task.

We define our state space $S = (u, \phi, q, \dot{q}, a_{t-1}, c_x, c_\omega, c_f, \hat{\mathbf{v}}, \hat{\mathbf{f}})$, where $u$ is the current RM state, $\phi$ is the number of time steps which occurred since the previous RM state changed, $q$ and $\dot{q}$ are the 12 joint angles and joint velocities respectively, $a_{t-1}$ is the previous action, $c_x$ and $c_\omega$ are base linear and angular velocity commands respectively, $c_f$ is the gait frequency command, and $\hat{\mathbf{v}}, \hat{\mathbf{f}}$ is estimated base velocity and foot heights. The RM state is encoded as a one-hot vector, making the dimensions of $S \in [49, 52]$ based on the number of RM states defining the gait.

**Gait Frequency:** Aside from gait specification, we also leverage RMs to specify gait frequency. Our definition of $\delta_r$ naturally encourages high frequency gaits, because maximizing the number of pose transitions maximizes total accumulated reward. Thus, we introduce gait frequency command $c_f$, which denotes the minimum number of environment steps which must be taken until the agent is allowed to transition to a new RM state. When the agent maximizes the number of RM transitions it takes, while being restricted by $c_f$, then the commanded gait frequency is followed. Adding $c_f$ on its own would cause the reward function to be non-

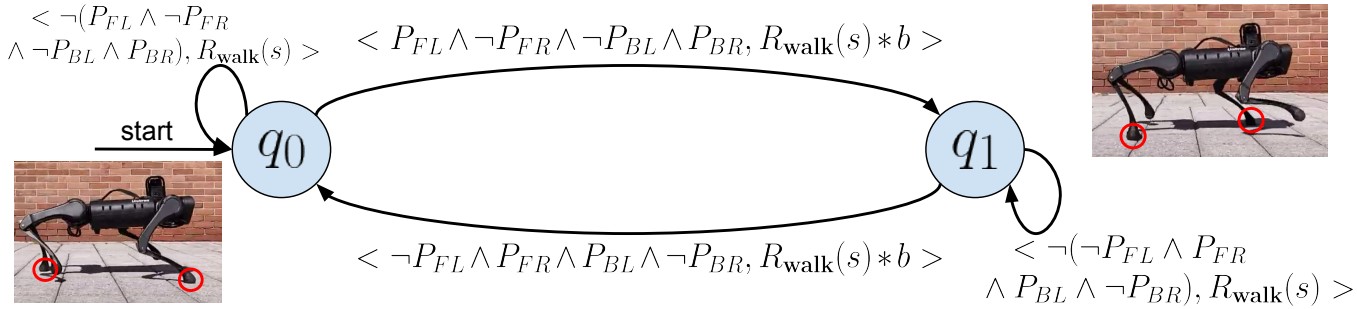

Figure 3: Reward Machine for **Trot** gait, where we want to synchronize lifting the FL leg with the BR leg, and the FR leg with the BL leg. **Trot** is one of the six gaits considered in this work.

Markovian, because the agent needs to remember how many environment steps have occurred since the RM state last changed. Thus, we also add timing variable $\phi$ to our observations, which keeps track of how many environment steps have occurred since the RM state has changed last. At every environment time step, we compare $\phi$ with $c_f$, and do not allow an RM transition to take place if $\phi < c_f$. Adding $c_f$ and $\phi$ to the state enables gait frequency to be dynamically adjusted during policy deployment, and is demonstrated on hardware in our supplementary video.

**Illustrative Gait:** We now discuss specifying a well known quadruped locomotion gait, **Trot**, via RM. Figure 3 shows the RM associated with this gait. In this **Trot** automaton, we want to synchronize lifting the FL leg with the BR leg, and the FR leg with the BL leg. LTL formula $P_{FL} \wedge \neg P_{FR} \wedge \neg P_{BL} \wedge P_{BR}$ evaluates to true when only the FR and BL feet are in the air simultaneously, while $\neg P_{FL} \wedge P_{FR} \wedge P_{BL} \wedge \neg P_{BR}$ evaluates to true when only the FL and BR feet are in the air simultaneously. The two RM states correspond to which combination of feet were previously in the air. If the agent is in state $q_1$, then $P_{FL} \wedge \neg P_{FR} \wedge \neg P_{BL} \wedge P_{BR}$ must have been evaluated as true at some point earlier. Note that when the agent does not achieve the desired pose, then the agent takes a self-loop to remain in the current RM state. [1]

**Remark** It is an intuitive idea of training a gait-specific locomotion policy via RM, because along with low-level sensor information, the policy also has access to the current RM state, which is an abstract representation of the historical foot contacts relevant to the current pose in the gait. Rather than attempting to learn this from a long history of world states, the RM state explicitly encodes the previously reached gait pose. Thus, the policy can learn different gaits in a sample-efficient manner, because at each time step it can reference the RM state to indicate which pose within the gait to reach next.

---

[1] We provide the RMs for all other gaits we trained in Supplementary Materials.

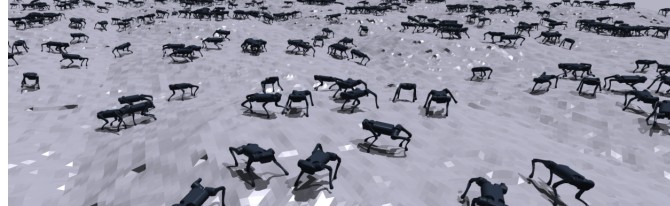

Figure 4: Isaac Gym simulation environment.

## 4 Experiments

We train six different locomotion gaits via RMLL in simulation, and perform an ablation study to evaluate whether knowledge of the RM state improves sample efficiency when compared to ablations which do not access the RM state during training. After that, we compare energy consumption and stability of each gait across different terrains. Finally, we demonstrate all learned gaits on a Unitree A1 robot.

### 4.1 Training Details

**State, Action, Reward** We estimate base velocity $\hat{\mathbf{v}}$ and foot heights $\hat{\mathbf{f}}$ concurrently with the policy, via supervised learning (Ji et al. 2022). Note that during training we only consider a foot in the air if it is higher than 0.03 meters. Actions include the target joint positions of each joint. These are input to a PD controller which computes the joint torques. The PD controller has a proportional gain $K_p = 20$ and derivative gain $K_d = 0.5$. The policy is queried at 50 Hz, and control signals are sent at 200 Hz. We set bonus $b = 1000$ in $\delta_r$ for all gaits.

**Environment Details** We use the Isaac Gym (Makoviychuk et al. 2021) physics simulator and build upon a legged locomotion environment (Rudin et al. 2022) to train our policies. We use a terrain called `random_uniform_terrain`, which includes hilly perturbations on the surface (see Figure 4). The robot traverses more challenging versions of this terrain based on a curriculum which increases terrain difficulty after the robot learns to traverse flatter versions of the terrain. Each episode lasts for 20 seconds, and ends early if the robot makes contact with the ground with anything other than a

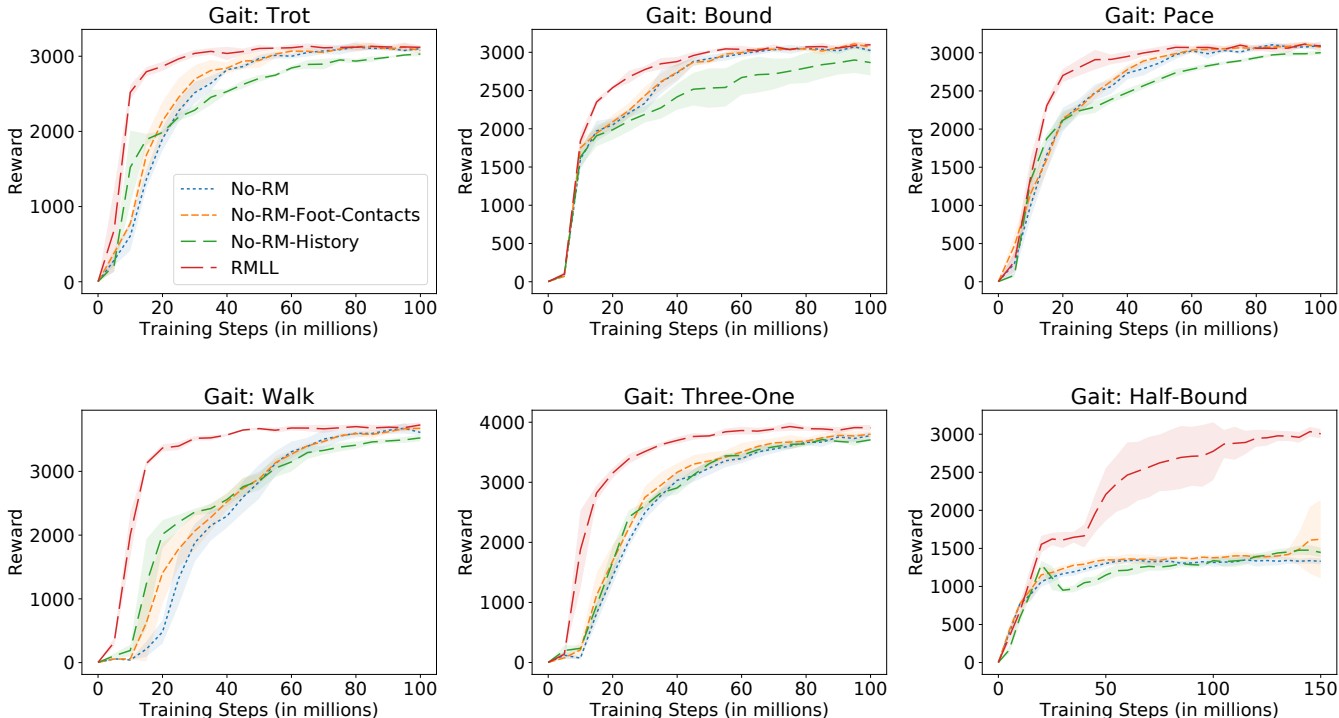

Figure 5: Reward curves for all gaits. RMLL more efficiently accumulates reward for each gait, particularly for the gaits with more complex foot contact sequences **Walk**, **Three-One**, and **Half-Bound**.

foot, if joint angle limits are exceeded, or if the base height goes below 0.25 meters. After each training episode, we sample a new velocity and gait frequency command for the robot to track. To facilitate sim-to-real transfer, we perform domain randomization over surface frictions, add external pushes, and add noise to observations (Rudin et al. 2022).

**Model Training**   We train our policy via PPO (Schulman et al. 2017), with actor and critic architectures as 3-layer multi-layer perceptrons (MLPs) with hidden layers of size 256. Each policy is trained for 100 million time steps (except for **Half-Bound**), where parameters are updated every 100,000 time steps. Data is collected from 4096 agents running simultaneously.

### 4.2   Ablation Study

We run an ablation study to determine whether knowledge of the RM state actually improves sample efficiency. We design the following baselines which we compare RMLL against:

1. **No-RM**: Remove the RM state from the state space, keeping everything else the same.
2. **No-RM-Foot-Contacts**: Remove the RM state from the state space, and add a boolean vector of foot contacts.
3. **No-RM-History**: Remove the RM state from the state space, and add a boolean vector of foot contacts. Expand the state space to include states from the past 12 time steps.

Comparing against **No-RM** indicates whether the RM state is useful at all. Comparing against

**No-RM-Foot-Contacts** indicates whether RM state is only useful because it contains information about foot contacts. Comparing against **No-RM-History** indicates whether the information provided by the RM state can be easily learned when given sufficient history.

It should be noted that there are no existing methods supporting the specification and learning of *arbitrary* gaits without using motion priors, dynamics models, or significant manual efforts such as prompt engineering and random pattern generators. Furthermore, highly customized gaits such as **Three-One** and **Half-Bound** are new to the literature, and to the best of our knowledge, there are no existing methods which support learning such gaits. Thus, we focus on evaluating how knowledge of the RM state contributes to the overall performance of RMLL.

We experiment over six different locomotion gaits: **Trot**, **Pace**, **Bound**, **Walk**, **Three-One**, and **Half-Bound**. See Supplementary Materials for the RMs defining each gait. Gaits **Trot**, **Bound**, **Pace**, **Three-One**, and **Half-Bound** sample linear and angular velocity commands from [-1, 1] meters per second, and a gait frequency command from [6, 12] time steps. Meanwhile, **Walk** samples from [-0.5, 0.5] meters per second and [5, 10] respectively. This is because quadruped animals naturally use **Walk** gait for slower locomotion speeds.

For each approach (ablation or not), we trained over five different random seeds per gait. For each training run, we save the policy after every 5 million steps. We then deploy each of those saved policies for 100 episodes, and average

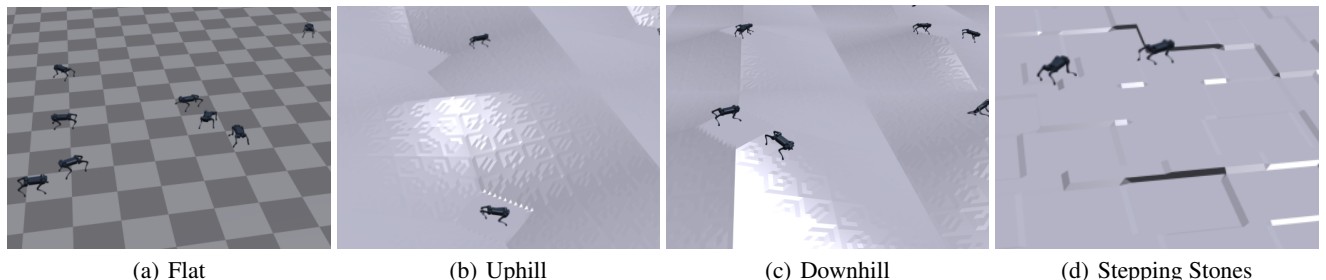

|            | (a) Flat | (b) Uphill | (c) Downhill | (d) Stepping Stones |

Figure 6: Visualization of each terrain we measured energy consumption and stability on.

| Gait | | Flat | Uphill | Uphill (Steep) | Downhill | Downhill (Steep) | Stepping Stones |
|------|---|------|--------|----------------|----------|------------------|-----------------|
| Trot | Energy | **2165.04** | **2628.88** | 5232.26 | **2137.38** | 2606.82 | **3149.81** |
|      | Stability | **0.00** | 1.49 | 5.51 | **0.10** | 6.35 | 2.30 |
| Bound | Energy | 4657.94 | 4962.13 | 5118.81 | 4091.22 | 3187.78 | 5099.17 |
|       | Stability | **0.00** | 3.09 | 5.70 | 2.77 | 6.79 | 2.16 |
| Pace | Energy | 2998.95 | 3023.70 | **3489.12** | 2811.13 | **2243.44** | 3493.30 |
|      | Stability | **0.00** | 1.65 | 3.09 | 0.69 | 5.02 | 1.33 |
| Walk | Energy | 4271.82 | 4828.25 | 6048.98 | 3747.29 | 2828.21 | 4354.12 |
|      | Stability | **0.00** | **0.70** | **2.73** | 0.56 | **4.54** | **1.28** |
| Three-One | Energy | 3145.81 | 3573.74 | 4118.70 | 3096.75 | 2509.71 | 3871.34 |
|           | Stability | **0.00** | 1.45 | 3.22 | 1.79 | 5.27 | 1.89 |
| Half-Bound | Energy | 4873.29 | 5568.86 | 5935.81 | 4399.62 | 3594.13 | 5308.95 |
|            | Stability | **0.00** | 3.56 | 7.87 | 3.18 | 6.76 | 3.25 |

Table 2: Comparing average energy consumption per meter (rows labelled as Energy) and stability (rows labelled as Stability) across different gaits and terrains.

the accumulated reward over the five runs per approach. We report the resulting reward curves in Figure 5, where the shaded region indicates the standard deviation of the total accumulated reward across the five training runs.

The results indicate that knowledge of the RM state improves sample efficiency for all gaits when compared with the ablations. We believe this is the case, because the RM state can efficiently inform the policy of gait-relevant historical foot contacts, whereas the ablations either do not have access to historical foot contacts, or must learn the relevant contacts from a history of world states.

The results also show that **No-RM-History** does not perform better than the other ablations without history, indicating that it is challenging to learn gait-relevant information directly from 12 time steps of historical states. We also notice that **No-RM** performs similarly to **No-RM-Foot-Contacts**, which indicates **No-RM** learns to implicitly estimate foot contacts from the state. Finally, we notice a large performance gap between RMLL and all other ablations for **Half-Bound**. We believe this is the case due to the additional complexity in the RM structure of this gait, which can be seen in Supplementary Materials.

### 4.3 Gait Comparisons

In order to further motivate learning different gaits, we compare energy consumption and stability for each gait across different terrains. We deploy each of our trained policies in

simulation, over four different types of terrains: {*Flat*, *Uphill*, *Downhill*, *Stepping Stones*} (See Figure 6). We also experiment with a version of *Uphill* and *Downhill* with steeper slopes, referred to as *Uphill (Steep)* and *Downhill (Steep)* respectively. Each policy (five policies over five random seeds per gait) is run for 20000 seconds per terrain type, and samples a base velocity command from [0.5, 1.0] meters/second ([0.25, 0.5] meters/second for **Walk**) and gait frequency command from the same range used during training. We measure energy consumption by multiplying motor torques by motor velocities, in the same manner as related work (Fu et al. 2021; Margolis and Agrawal 2023). We consider the robot to have fallen over when the base touches the ground, and reset the episode on a fall. Results are reported in Table 2. We find that **Trot** consumes the least energy on most terrains, although **Pace** consumes the least energy on steeper slopes. Meanwhile, **Walk** is the most stable on all terrains except for *Downhill*, where **Trot** is the most stable.

### 4.4 Qualitative Results

**Foot Contacts** In order to better highlight the behavior of each of the six gaits we learned, we report foot contact plots for each gait. In simulation, we deploy each of our six gaits at a constant linear velocity, while setting $c_f = 6$ for all gaits. We record the foot contacts of each gait in Figure 7, which shows that each of our gaits follows the expected foot contact sequence and gait frequency. For example, green and

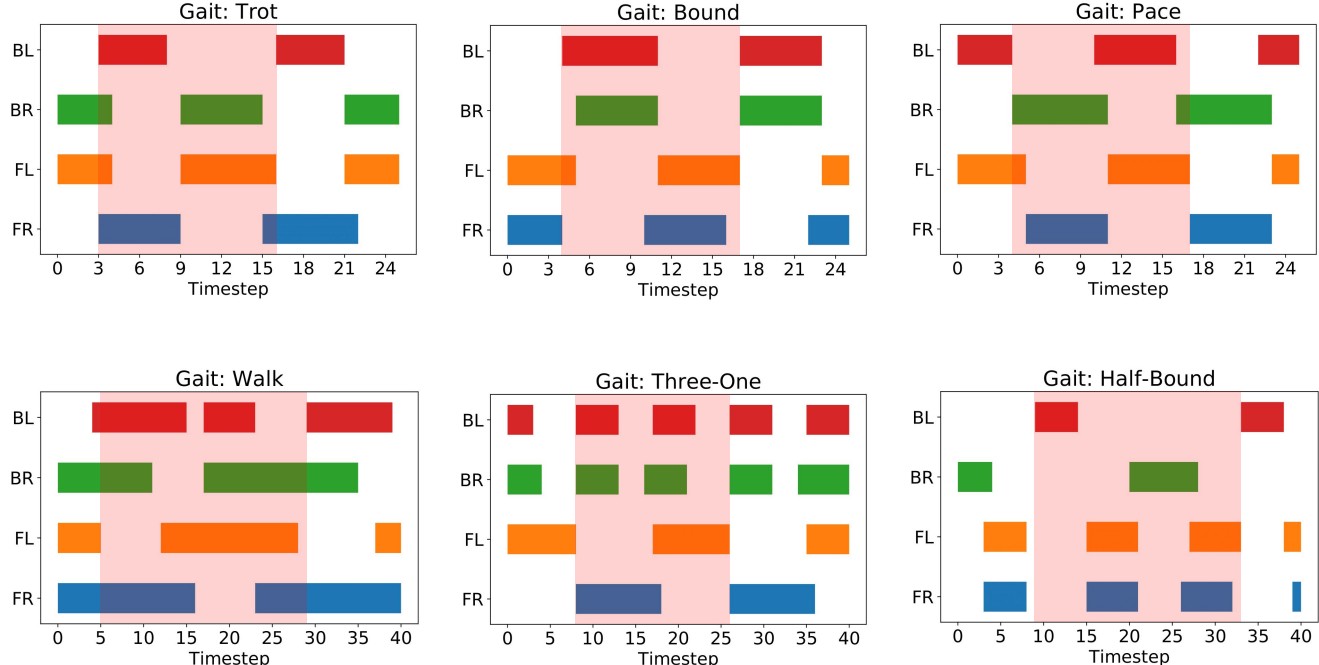

Figure 7: Foot contact plots for each gait learned using RMLL. We report foot contacts from trials in simulation where we deploy each gait with a constant forward velocity command, zero angular velocity command, and constant gait frequency command. The colored horizontal bars indicate when the specified foot makes contact with the ground. The red highlighted region in each subplot denotes a full cycle of the gait.

orange bars in **Trot** are synchronized, indicating BR/FL feet are coordinated.

**Hardware Demonstration**   We transfer our learned policies from simulation to a real Unitree A1 robot, without any additional fine-tuning. Each trial is on an outdoor concrete walkway, where we increase and decrease gait frequency throughout the trial. The robot is sent velocity commands in real time via a joystick, operated by a human. We find that RMLL policies from all gaits successfully transfer to hardware, and the intended foot contact sequence is realized. We also find the robot is capable of dynamically adjusting gait frequency according to the gait frequency command. A video capturing each of these trials is included in the supplementary video.[2]

## 5   Discussion

**Limitations and Future Work**   While our approach can be used to easily specify and learn customized locomotion gaits and gait frequencies, we have not studied how to optimally leverage these different gaits to efficiently traverse various terrains, nor have we studied how to smoothly transition between gaits. In future work, researchers can train a wide variety of gaits, and learn how and when to transition between gaits and gait frequencies to most quickly or efficiently traverse different terrains. This can involve a

two-stage procedure which trains a high-level policy to select pre-trained RM policies based on current terrain conditions, or an end-to-end approach which concurrently trains RM-based locomotion policies and a gait selector. Another interesting direction is to specify and dynamically adjust other gait parameters such as stride length, base height, or foot height. Finally, another avenue for future research is to investigate extracting descriptions of novel gaits from pre-trained large language models (Tang et al. 2023; Yu et al. 2023), and to convert the descriptions to formal representations from which RL agents can learn locomotion policies.

**Conclusion**   We leverage reward machines to easily learn and specify different quadruped locomotion gaits and gait frequencies. This is done via automatons over simple logical rules which specify desired foot contact sequences. We efficiently train locomotion policies in simulation which learn these specified gaits over a range of gait frequencies, without the use of motion priors, dynamics models, or significant manual efforts when compared to existing work. We evaluate the sample efficiency of our approach in simulation through ablation studies. We also evaluate the stability and energy consumption of each of our learned gaits over varying terrains. Finally, we demonstrate policies of each of our six gaits on hardware, and find that our robot can perform a variety of different gaits, while dynamically adjusting gait frequency.

---

[2]Supplementary materials can be seen at our anonymous site: https://sites.google.com/view/rm-locomotion-learning/home

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
