# OpenReview forum: "Learning Quadruped Locomotion Policies using Logical Rules"
_icaps-conference.org/ICAPS/2024/Conference — ICAPS 2024_

### Official Review · Reviewer_9giq · 2024-01-20

**Significance And Importance:** 2
**Soundness:** 2
**Novelty:** 2
**Clarity:** 3
**Overall Evaluation:** 1
**Confidence:** 3

**Weaknesses:**

1: Minor weaknesses that are easily fixable.

**Contributions Of The Paper:**

The contribution of the paper consists of a method, called RM-based Locomotion Learning (RMLL), to learn six different kinds of gaits for a quadruped robot based on high-level logic rules describing the gaits rather than reference trajectories and the robot’s motion constraints.
Such an approach has been evaluated both in simulation and on a real robot.

**Ethical Considerations:**

(1) Not Applicable: The paper does not have any ethical considerations to address

**Nomination For Best Paper:**

No

**Questions For Authors:**

(1) Figure 3 shows the Reward machine for trot gait. Other reward machines are defined for the six explored gaits as reported in the Supplementary materials. How the kind of gait is handled/inputted by/in the system?
(2) Could the agent learn to combine multiple gaits sequentially?
(3) Thus, the policy can learn different gaits in a sample-efficient manner, because at each time step it can reference the RM state to indicate which pose within the gait
to reach next. → please provide further details related to how this sort of “prediction” is implemented.
(4) Beyond the reward, please report the loss function too.
(5) How do you compute the energy and stability reported in Table 2?

**Reproducibility:**

1: Difficult to reproduce because of missing detail.

**Strengths Of The Paper:**

One of the strengths of this approach is the possibility of representing the different gaits via logical rules without using motion priors.
Furthermore, the paper is well-written. The authors greatly introduce the similarities and differences of the proposed methods with respect to the other state-of-the-art approaches.
Finally, the approach is evaluated in a real-life experiment, which has been shown in a video.

**Weaknesses Of The Paper:**

There are too many references to the content reported in the Supplementary material.
In addition, the work is difficult to reproduce. Please report further technical details about the hardware used to train the policies. The link to the code would be useful.
Finally, please see the questions reported in the next point.

---

> ### Author Rebuttal · Authors · 2024-01-27
>
> We thank the reviewer for the detailed review.
>
> Response to **Questions For Authors:**
>
> >1. … How the kind of gait is handled/inputted by/in the system?
>
> Each gait was manually specified in the form of an automaton (Figure 3 as an example). The automaton was then used for generating the reward function that was used for providing feedback to the learning agent in simulation. As a result, we were able to train a different policy per gait. Finally, the learned policies were directly transferred to the real robot. Running different policies would demonstrate different gaits. We will update the *Hardware Demonstration* paragraph in Section 4 for better clarity.
>
> >2. Could the agent learn to combine multiple gaits sequentially?
>
> We discussed it in Line 487. It’s a very interesting problem, which we leave for future work.
>
> >3. …please provide further details related to how this sort of "prediction” is implemented.
>
> We do not **explicitly** implement anything which can predict the next desired foot contacts. Because the RM state encodes the gait-relevant foot contacts, the agent can leverage this representation to learn what the next foot contacts should be. The agent is encouraged to learn this, due to the reward bonus provided by taking RM transitions. In other words, the agent automatically and efficiently learns the important foot contact sequences during training when given knowledge of the RM state.
>
> >4. Beyond the reward, please report the loss function too.
>
> We use the loss function defined by the commonly used PPO algorithm (see algorithm 1 in Schulman et al. 2017), which was cited in this paper in Line 366. In the next version, we will add a line right after citing the PPO algorithm to clarify.
>
> >5. How do you compute the energy and stability reported in Table 2?
>
> We mention in line 450 how we compute energy consumption: *"We measure energy consumption by multiplying motor torques by motor velocities, in the same manner as related work (Fu et al. 2021; Margolis and Agrawal 2023)."*
>
> Stability is reported as the average number of falls over all random seeds per gait. We will update Section 4.3 to include a description about how stability is computed. Thanks for pointing it out!
>
> Other than responding to the above questions, we will address the weaknesses about references and source code. In particular, we added a link to our anonymized source code on the project website: https://sites.google.com/view/rm-locomotion-learning/home

---

### Official Review · Reviewer_K8Ft · 2024-01-21

**Significance And Importance:** 2
**Soundness:** 3
**Novelty:** 3
**Clarity:** 4
**Overall Evaluation:** 2
**Confidence:** 3

**Weaknesses:**

1: Minor weaknesses that are easily fixable.

**Contributions Of The Paper:**

Paper presents a novel and interesting approach to efficiently learning robot gaits using a Reward Machine for gait specifications.  Extensive simulation experiments demonstrate the impact of the model, and the learned policies transfer to real hardware.

**Ethical Considerations:**

(1) Not Applicable: The paper does not have any ethical considerations to address

**Nomination For Best Paper:**

No

**Questions For Authors:**

If your models have no accepting states, what is the purpose of gait M in Figure 2, where state E is a sink state.

In you ablation studies - if you have removed RM, and thus knowledge of the state transition, how is the bonus applied?

The gait example you provide in the paper is purely propositional - there are no modal operators that are associated with LTL.  What is the purpose of using LTL, if it is really not being used?

The gait comparisons section is interesting, but would be even more so if you could compare against animal gaits.

**Reproducibility:**

3: Authors describe the implementation and domains in sufficient detail.

**Strengths Of The Paper:**

The overall approach is quite interesting.  The related work is very well done.  The paper is well written and the approach is well described.  The simulation experiments are extensive and seem convincing.

**Weaknesses Of The Paper:**

Lots of parameters, and unclear how sensitive the approach is to them.  In particular, the bonus term is very large (1000), which would seem to dominate all the other rewards - how sensitive is the approach to that value?  Also, where did the terms of R_walk come from - why were the terms chosen, why are they defined as they are (e..g, linear x velocity is an exponential, while linear z velocity is quadratic), how were the scaling factors chosen, and how senistive is the learning to all these choices?

---

> ### Author Rebuttal · Authors · 2024-01-27
>
> We thank the reviewer for the detailed review.
>
> Response to **Questions For Authors:**
>
> >If your models have no accepting states, what is the purpose of gait M in Figure 2, where state E is a sink state.
>
> The gaits shown in Figure 2 are for illustrating the diversity of gaits. We didn’t mean to use them to refer to any specific gaits.
>
> It’s indeed the case that the RM definition includes a set of accepting states, as stated in Line 231 in the paper. However, the set of accepting states of RMLL (ours) is always empty, as stated in Line 270, because locomotion is an infinite-horizon task. In the next version, we will update the automaton of Gait M to avoid the sink states for better clarity.
>
> >... if you have removed RM, and thus knowledge of the state transition, how is the bonus applied?
>
> In all ablations, we still kept track of what the RM state was. However, the knowledge about the RM states was not exposed to the baseline agents. Thus, we could still compute the bonus for all agents (baselines and ours) in a fair way, and evaluate whether the RM state knowledge was useful for the agents or not.
>
> >The gait example you provide in the paper is purely propositional … What is the purpose of using LTL, if it is really not being used?
>
> RMLL (ours) was built on existing research on RMs. One can use LTL formulas to construct RMs or directly encode the automatons to do so. In this paper, we presented our constructed automatons for better clarity, though those RMs could have been constructed using LTL formulas.
>
> It was our mistake to mention using LTL to build RMs in this research. We will avoid such statements in the next version (in the caption of Figure 2 and in the *Illustrative Gait* in Line 308). We greatly appreciate the comment.
>
> >The gait comparisons section is interesting, but would be even more so if you could compare against animal gaits.
>
> If the reviewer was suggesting comparisons against gaits learned from data collected from animals (say dogs), then we agree that this can lead to interesting experiments. There exist publicly available datasets (in the form of reference trajectories) collected on dogs as cited in our submission (Peng et al. 2020). Potentially, the experiment results can be used for further justifying the effectiveness of our RMLL-based gait specification approach.
>
> We additionally respond to the stated weaknesses by noting that R_walk came from Rudin et al. 2022, and bonus sensitivity ablations are left for future work.

---

### Official Review · Reviewer_EcEz · 2024-01-26

**Significance And Importance:** 2
**Soundness:** 3
**Novelty:** 3
**Clarity:** 4
**Overall Evaluation:** 2
**Confidence:** 4

**Weaknesses:**

1: Minor weaknesses that are easily fixable.

**Contributions Of The Paper:**

The authors presented in this work a solution for reinforcement learning of quadrupedal locomotion gait by utilising reward machines and logical specifications of different gaits. The approach is shown for several hand-crafted gait specifications based on the foot contact. The approach was compared to a naive approach of reinforcement learning without having access to the state of the reward machine. Learned policy was also demonstrated on the real quadrupedal robot.

**Ethical Considerations:**

(5) Excellent: The paper comprehensively addresses all of the applicable ethical considerations

**Nomination For Best Paper:**

No

**Questions For Authors:**

-Can you explain how would this method scale to other problems?
-Can you elaborate more on the comparison with the naive RL approach?

**Reproducibility:**

3: Authors describe the implementation and domains in sufficient detail.

**Strengths Of The Paper:**

-The paper is well-written, and clearly presented and the main contributions are highlighted.
-Related work is relatively well presented and the whole idea is well-motivated.
-The presented idea is extensively validated and the performance is promising.
-Presented approach seems to be useful for learning locomotion gait

**Weaknesses Of The Paper:**

-The approach seems to be focused on a locomotion problem. It a bit unclear how this approach would generalise to other problems, perhaps this aspect should be discussed further.
-The comparison with naive RL that does not have access to the state of the RM seems unfair as the problem is then non-Markovian.

minor:

-line 70, specify domains.
-line 137, remove “with motion priors”->”motion priors”
-line 240, does really agent evaluate automaton state?
-R_{walk} is not explained anywhere. Same goes for “b”

---

> ### Author Rebuttal · Authors · 2024-01-27
>
> We thank the reviewer for the detailed review.
>
> Response to **Questions For Authors**:
>
> > -Can you explain how would this method scale to other problems?
>
> First of all, we believe gait learning is a very important problem in robotics. RMLL (ours) can directly scale to other non-quadruped legged robots (such as bipeds and hexapods) after adding or removing necessary propositional symbols, one for each foot. To further scale to learning complex behaviors, the implementation of RMLL would require the perceptual capability of robot poses, e.g., using motion capture systems. That’s a very interesting direction for future research.
>
> > -Can you elaborate more on the comparison with the naive RL approach?
>
> We believe this question is a follow-up of the point made in *Weaknesses Of The Paper*, where it was stated that *"The comparison with naive RL that does not have access to the state of the RM seems unfair as the problem is then non-Markovian.”*
>
> Although the different agents (including RMLL) in experiments were exposed to different state information, they all lived in the same world and were evaluated using the same reward function. For example, in the “bound” gait, both RMLL and No-RM agents received the same bonus reward for synchronizing the front feet, though the No-RM agent was not aware of the high-level state transition when they were synchronized.

---

### Meta-Review · Area_Chair_kJMY · 2024-02-06

**Recommendation:** Accept (Oral)
**Confidence:** 5

**Metareview:**

The paper proposes a novel approach for using reward machines (RMs) to learn quadruped locomotion policies. The paper breaks down each step; all the necessary variables; describes the gates and their rationale; and so on. Experiments illustrate the results of the RM approach for six gates, with metrics such as energy and stability, across six different terrains. They also explore differences in foot contact points and provide visual evidence in 3d simulation.

The reviewers praised the paper on a number of points, such as the high-quality writing, related work, problem description, presented approach, experimental results, so on. There a few points of clarification, such as the challenge to reproduce the results and the large number of parameters to learn, which were mostly clarified in the rebuttal.

Overall, the paper is a solid approach for learning quadruped locomotion, and does a great job presenting the problem and demonstrating its claims. We hope that the reviewers' feedback further refines the already great paper.

**Ethical Considerations:**

(1) Not Applicable: The paper does not have any ethical considerations to address